# Genomic Analyses Implicate the Amazon–Orinoco Plume as the Driver of Cryptic Speciation in a Swimming Crab

**DOI:** 10.3390/genes13122263

**Published:** 2022-12-01

**Authors:** Pedro A. Peres, Heather Bracken-Grissom, Laura E. Timm, Fernando L. Mantelatto

**Affiliations:** 1Department of Biology, Institute of Environment, Florida International University (FIU), Miami, FL 33199, USA; 2Laboratory of Bioecology and Systematics of Crustaceans (LBSC), Faculty of Philosophy, Sciences and Letters at Ribeirão Preto (FFCLRP), University of São Paulo (USP), Ribeirão Preto 14040-901, Brazil; 3Department of Invertebrate Zoology, National Museum of Natural History-Smithsonian, Washington, WA 20013-7012, USA; 4College of Fisheries and Ocean Sciences, University of Alaska Fairbanks, Fairbanks, AK 99775, USA; 5Auke Bay Laboratories, Alaska Fisheries Science Center, NOAA National Marine Fisheries Service, Juneau, AK 99801, USA

**Keywords:** Brachyura, ddRAD-seq, diversification, hybridization, marine invertebrate, marine barrier, secondary contact, speciation with gene flow, swimming crab

## Abstract

The Amazon–Orinoco plume (AOP) is the world’s largest freshwater and sediment discharge into the ocean. Previous studies limited to mtDNA suggest that the swimming crab *Callinectes ornatus* Ordway, 1863 exists as two distinct genetic clusters separated by the AOP. However, questions concerning migration, diversification time, and species delimitation are unresolved. Densely sampling markers across the genome (SNPs) could elucidate the evolutionary processes within this species. Here, we combined mtDNA data and ddRAD-seq to explore the diversification patterns and processes within the swimming crab *C. ornatus*. We show great genetic differentiation between groups on the north and south sides of the plume but also signs of hybridization. Demographic modeling indicates the divergence between groups starting around 8 Mya following the AOP’s formation. After a period of isolation, we detect two incidences of secondary contact with stronger migration in concordance with the North Brazil Current flow. Our results suggest speciation with gene flow explained by the interplay among the AOP, oceanographic currents, and long larval dispersal. This work represents the first investigation employing ddRAD-seq in a marine invertebrate species with distribution encompassing the north and south Atlantic and sheds light on the role of the AOP in the diversification of a marine species.

## 1. Introduction

Understanding marine species diversification remains challenging for ecologists and evolutionary biologists [1]. The interplay among complex life cycles (including multiple larval stages), pelagic larval duration (PLD), and oceanographic processes results in convoluted diversity patterns in marine systems (e.g., [2,3]). Moreover, processes of diversification in the marine system can be easily overlooked because many marine barriers are not conspicuous (e.g., mountains in a terrestrial environment) [4]. Most marine barriers are “soft”, that is, their permeability depends on the species, representing physical isolation for some species or environmental gradients for others [4]. Therefore, it is hard to predict the effects of marine barriers on diversification at the intra- or interspecific level. There is evidence for low or high genetic structure, allopatric and sympatric speciation, speciation with gene flow, and cryptic speciation scenarios [3,5,6,7,8,9,10,11]. However, the mechanisms responsible for each response are usually unknown and local/taxa-dependent. By investigating the effects of soft barriers on marine taxa, advancements can be made toward understanding marine diversification.

One of the most notable marine barriers is the Amazon–Orinoco plume (AOP) [12]. Located in the tropical western Atlantic, the AOP is delimited by the Amazon and Orinoco river mouths in the northern part of South America and represents the world’s largest freshwater and sediment discharge into the ocean [12]. The AOP’s water outflow is 6300 km^3^/year and extends 400 km off the coast and 30 m in depth [13]. This represents an environmental barrier known to change water salinity (36 to 32 psu) and turbidity (silicate from 1 to 9 μmol/kg) [13,14]. The AOP originated approximately 8–10 million years ago (Mya) following the Andes uplift and continued to develop by intensifying river outflow and sedimentation until 2–5 Mya when it was fully established [15]. In some species, results from few molecular markers show evidence for genetic structure and sister species separated by the AOP; however, some other specific marine groups are not affected by this barrier (annelids—[16]; crustacean—[17]; fish—[18,19,20]; manatees—[21]; mussel—[22]). The factors driving each scenario are unclear, but we can hypothesize that low tolerance to freshwater and/or turbid water may be driving diversification patterns. For instance, there is evidence for a widespread western Atlantic coral species showing genetic groups separated by the AOP [23] and evidence of coral larvae being affected by salinity changes [24]. Moreover, the comparison of two swimming crab species with known distinct physiological salinity tolerances shows that they are affected in different ways by the AOP [25]. Unfortunately, salinity tolerance data are not available for many species, and the effect of the AOP upon diversification is assumed a posteriori.

When investigating genetic structure in marine organisms, we must also consider the effect of pelagic larval duration (PLD). Many marine species show indirect development with larval stages during their life cycle. The larvae are released in the water and have the potential to be transported by ocean currents and reach distant locations [26]. There is a positive association between the PLD and the distance traveled by the larvae [27]. Consequently, longer PLD would theoretically result in a lack of population differentiation (e.g., [8]). However, across marine species, the association between PLD and population differentiation has been shown to range from weak to null [28]. Indeed, many species show some level of genetic structure across their distribution [3,6].

Marine populations or species separated by the AOP represent promising candidates for investigations on the interplay between PLD and marine barriers because they can help us to understand if one factor can overpower the other. Studies analyzing species occurring on both sides of the AOP have only used a few markers (Sanger sequencing)—usually mitochondrial DNA (mtDNA) (e.g., [16,17,18,19,20,22]). There are examples of mtDNA not capturing structure found by loci across the nuclear genome (single nucleotide polymorphisms—SNPs; [29]), as well as cases in which mtDNA delineates several cryptic species that are not supported by SNP data [30], an event called mitonuclear discordance [31]. Thus, studies employing genomic methods examining more loci across the nuclear genome (single nucleotide polymorphisms—SNPs) are warranted (but see [29] and [32]) because genetic structure and speciation can be overlooked in species occurring on both sides of the AOP [33,34]. Some topics that demand further investigation are the divergence time between genetic groups separated by the AOP estimated by hundreds of loci, comparisons between results coming from mtDNA (Sanger sequencing) and genomic approaches, and differentiation across the genome.

The swimming crab *Callinectes ornatus* Ordway, 1863 represents the ideal model organism to investigate the effects of a soft barrier (AOP) on the diversification of marine species in the tropical western Atlantic. The species is widespread along the western Atlantic, occurring in coastal waters (up to 75 m depth) from south Brazil to North Carolina, USA [35,36]. Despite nine larval stages and a PLD of 50 days [37,38,39], the species shows a strong genetic structure, composed of two separated groups: one north and one south of the AOP [25]. Unlike many species for which physiological data are not available, there is evidence showing that *C. ornatus* is sensitive to salinity changes [40]. Considering this *a priori* data showing that *C. ornatus* is not tolerant to low salinities, the authors hypothesized that salinity tolerance plays a significant role in shaping these two groups [25]. Unfortunately, due to the limited resolution of the data (mtDNA—*cytochrome oxidase I* (COI) and *16S ribosomal RNA* (16S rRNA)), questions of demographic and diversification processes remain to be addressed. For instance, there is no clear resolution if two groups represent different species, and the results might represent only the mtDNA evolutionary history and not the species tree [31]. Therefore, using more loci can improve our characterization of these lineages, shed light on the role of the AOP upon western Atlantic species, and determine if *C. ornatus* represents a case of cryptic speciation.

Here, we combined mtDNA (COI gene) and ddRAD-seq to explore the genetic structure and demographic history of *C. ornatus*, focusing on the role of the AOP in the diversification within this species. We quantified genetic differentiation and investigated demographic processes between groups on both sides of the AOP (north and south tropical western Atlantic). Considering the scenario shown by mtDNA [25], we hypothesize that we will find evidence of speciation driven by the AOP when examining SNPs from across the genome and evidence that shared mtDNA haplotypes are the result of ancient polymorphisms. To the best of our knowledge, our work represents the first investigation employing ddRAD-seq in a tropical marine invertebrate species with distribution encompassing both sides of the AOP. We provide empirical evidence of a dynamic demographic history indicating that the AOP has led to diversification in a species displaying long PLD but low salinity tolerance.

## 2. Materials and Methods

### 2.1. Sampling and DNA Extraction for mtDNA and ddRAD-Seq

We used the same cytochrome c oxidase subunit I (COI) dataset for the mtDNA analyses as in [25]. We expanded it with sequences available on GenBank, plus de novo sequences generated and submitted to GenBank from regions not included in previous analyses (northern Brazil, French Guiana (region within the plume), North Carolina (USA), *n* = 12) (Appendix A). For the ddRAD-seq analyses, we obtained 63 individuals of *C. ornatus* (Figure 1) from the following collections: Crustacean Collection of the Department of Biology (CCDB)—Faculty of Philosophy, Sciences, and Letters at Ribeirão Preto (FFCLRP) of the University of São Paulo (USP); the Invertebrate Zoology Collection—Florida Museum of Natural History (FLMNH) of the University of Florida; and the Florida International University Crustacean Collection (FICC). Many of them were used for both mtDNA and ddRAD analyses, which enabled us to compare the results from both types of markers. Our sampling covers the species’ described range (Figure 2A and Appendix A).

Genomic DNA was extracted from muscle tissues using the salt extraction method [41,42] or with the DNeasy Blood and Tissue Kit (Qiagen, Hilden, Germany), following the protocol provided by the manufacturer.

### 2.2. COI Amplification and Analyses

COI sequences were amplified using the primer COL6b (5′-ACAAATCATAAAGATATYGG-3′)/COH6 (5′-TADACTTCDGGRTGDCCAAARAAYCA-3′) [43], following the PCR cycle: initial denaturing for 5min at 94–95 °C; annealing for 35–40 cycles: 45 s at 95 °C, 45 s at 42–48 °C, 1 min at 72 °C; final extension of 5 min at 72 °C. PCR products were purified using a SureClean Plus kit (Bioline Reagents, UK) following the protocol provided by the manufacturer, and sequenced in an ABI 3730 xl DNA Analyzer (Applied Biosystems, Waltham, MA, USA) following Applied Biosystems protocols. Primer removal and quality checks were performed in Geneious Prime 2020.2.4 (https://www.geneious.com). Pseudogenes were identified by translating the consensus sequences and checking for indels and stop codons, removing them when present [44]. We aligned the sequences using MAFFT v.7 [45] and prepared a haplotype file in DnaSP v.6 [46], which was used to access the relationship among haplotypes using a statistical parsimony network through the TCS method [47] implemented in PopArt v.1.7 [48].

### 2.3. ddRAD-Seq Library Preparation and Data Processing

Double digest RADseq libraries were prepared according to the ddRADseq method [49]. Briefly, after enzyme trials to determine the best enzyme set, DNA from all individuals was digested with a combination of *NlaIII* and *NotI* (New England Biolabs, Ipswich, MA, USA). Following digestion, custom barcoded adapters were ligated to the fragments and pooled into twelve sublibraries. Each sublibrary was size selected (250–300 bp) on a PippinPrep with a 2% Agarose Gel Cassette (SageScience, Beverly, MA, USA). Size-selected fragments were then amplified via PCR with Phusion Hi-Fidelity Polymerase (Thermo Scientific, Waltham, MA, USA), which also incorporated indices (i7) and Illumina adapters into the fragments, allowing for the pooling of sublibraries into the final library. A washing step using AMPure XP beads (Beckman Coulter, Brea, CA, USA) was performed after digestion, adapter ligation, and PCR. Sequencing was done at the Genewiz Facility in South Plainfield, New Jersey, by an Illumina HiSeq 4000 (paired-end [PE] 150).

Raw sequence files were demultiplexed (--inline-index), cleaned (-c), quality-filtered (q), and rescued (-r), allowing two mismatches using the *process_radtags* program in STACKS v.2.3d [50]. Short reads were indexed and aligned to the blue crab (*Callinectes sapidus*) reference genome (Csap_IMET_V1, [51]) using Burrow–Wheller Aligner (BWA-MEM) v. 0.7.17 [52,53,54] with default settings. Generated SAM files were converted to BAM files using SAMtools v.1.9 [55]. Using BAM files as input, we ran the ref_map.pl wrapper program on STACKS v.2.3.d to call SNPs. We applied an iterative filtering strategy to minimize the effects of missing data and low-quality sequencing when calling SNPs. Using STACKS v.2.3d, we first ran the *populations* module to detect genotype call rate per locus (i.e., the proportion of individuals a locus is called in a given population; -r) using no threshold. Then, we used the function --missing-indv on VCFtools v. 0.1.17 [56] to assess missing data per individual and excluded individuals with >95% missing data. A second round excluded loci with a call rate <25% (-r 0.25) and individuals with >85% missing data. A third round was used to retrieve all loci with a genotype call rate of at least 50% (-r 0.5). This strategy is effective to exclude low-quality loci and individuals [57]. We set a minor allele frequency of 5% (--min-maf 0.05) because low-frequency alleles can affect population structure inferences [58]. The population module was also used to filter one SNP per rad-loci (--write-random-snp).

### 2.4. Linkage-Disequilibrium and Outlier Detection

Because most of the analyses assume that SNPs are independent, SNPs potentially under linkage disequilibrium (LD) were detected by calculating the squared correlation coefficient (r^2^) between SNP pairs using the function --geno-r2 in VCFtools v. 0.1.17. Only pairwise SNP comparisons within the same chromosome and with more than half of our final sample size were considered. SNPs showing r^2^ > 0.8 were excluded from the final dataset. Two different approaches were implemented to identify outlier SNPs (i.e., non-neutral SNPs, possibly under selection), BayeScan v.2.1 [59] and PCAdapt [60], as the use of multiple methods has been recommended to reduce type 1 errors [61,62,63]. BayeScan is a method to identify putative adaptive SNPs based on different allele frequencies among populations, and we performed it using default settings (prior odds to 10, iterations to 5000, and burn-in to 50,000). Outlier SNPs were identified at a q-value (i.e., false discovery rate) of 0.01. PCAdapt implements a hierarchical method (not assuming groups a priori) based on principal component analysis that identifies SNPs excessively related to population structure, probably due to selection. We ran PCAdapt exploring twenty PCs (K = 20) to select the optimal K following Cattell’s rule to retain the best K PC value [60]. These PCs detect the SNPs most associated with population structure. We detected putative non-neutral SNPs based on a q-value of 0.01. An SNP was considered non-neutral when both analyses indicated that the SNP is potentially non-neutral.

### 2.5. Genetic Diversity, Population Structure, and Genetic Differentiation

Summary genetic diversity statistics were calculated in GENODIVE v3.0 [64]. We employed the Bayesian program STRUCTURE v2.3.4 [65] to test for population structure within the data. Eight K-values were tested (K = 8) 10 times each under the admixture model. Following a burn-in of 10,000 generations, 100,000 Markov Chain Monte Carlo generations were run. The most optimal K value was estimated with the posterior probability for each model using the ΔK method [65,66], as implemented in STRUCTURE HARVESTER v0.6.94 [67]. We calculated corrected pairwise F_ST_ in GENODIVE v3.0 between the groups in the north and south of the AOP. We also accessed genetic differentiation across the genome by calculating kernel-smoothed F_ST_ considering a sliding window of 900 Kbp in total length using the population module in STACKS v.2.3d.

### 2.6. Demographic History: Divergence Time, Migration, and Effective Population Size

We performed demographic modeling analyses to find the best scenario explaining the two genetic groups—north and south, separated by the AOP (see results). Simulated and observed two-dimensional joint site frequency spectra (2D-JSFS) were compared using the Genetic Algorithm for Demographic Model Analysis (GADMA, [68]). Loci were considered not linked. Populations were projected down to smaller sample sizes (projection parameter set to north: 9 and south: 44) for 2D-JSFS estimation. Ordinary differential equations implemented in the *moments* engine [69] were used to simulate 2D-JSFS within GADMA. The Genetic Algorithm (GA) was used for global optimization and Powell’s conjugate method was used for local optimization during the model search in the parameter space. Optimizations were run for 50 repeats for each model. We set effective population size changes and migration to “true”, no demographic changes before the first split were allowed, and two demographic changes after the first split were allowed (initial structure: [1,1]; final structure [1,2]). All other parameters were set to default. Models were compared using Akaike’s Information Criterion (AIC). Parameters are given in genetic units in relation to reference effective population size (N_ref_). N_ref_ was determined using θ/θ_0_, in which θ is the population-scaled mutation rate given in the output of the GADMA run and θ_0_ = 4 × µ (mutation rate per base per generation) × L (length of the sequence that was used to build the data). Because there is no µ estimate for *C. ornatus* or swimming crabs, we used the value calculated for *Drosophila melanogaster* [70] and *Daphnia pulex* [71]: 4 × 10^−9^. The average sequence length of the ddRAD-loci used as L = 188.41 bp. *Callinectes ornatus* generation time was considered as one year [72].

## 3. Results

### 3.1. mtDNA (COI)

We used 66 sequences (570 bp), including 12 new sequences not included in [25], representing North Carolina, northern Brazil, and French Guiana (within the AOP). The haplotype network depicts a clear split between the individuals from the north and south of the AOP, but one individual from the north group (FLMNH 32103) falls within the south network. Individuals from the AOP region show the most common haplotype, shared by the south group (Figure 2B).

### 3.2. ddRAD-Seq

After process_radtags (demultiplexing, cleaning, and quality-filtering), we retained ~570 million reads across 63 individuals (69% retained reads). After the ref_map pipeline, the unfiltered set of markers consisted of 91,098 loci, composed of 15,501,392 sites, covering approximately 2% of the genome. After the filtering steps, 10 individuals were excluded (16% of our total) and we kept 53 individuals (Appendix A). The ref_map pipeline returned 1870 loci (mean = 188.41 bp) and 912 SNPs (one SNP per rad-locus).

### 3.3. Linkage-Disequilibrium and Outlier Detection

Thirteen SNPs showed signals of LD and were excluded from the dataset. Although BayeScan and PCAdapt detected potentially non-neutral SNPs, the analyses did not converge to the same SNP set. Therefore, we decided not to exclude any SNPs and assume differences in allele frequency and genetic differentiation in some loci due to neutral processes (e.g., genetic drift). Our final dataset for downstream analyses was composed of 899 SNPs. 

### 3.4. Genetic Diversity, Population Structure, and Genetic Differentiation

Summary statistics of genetic diversity for the north and south groups are similar and are shown in Table 1. STRUCTURE results show K = 2 as the most probable number of genetic clusters within our dataset. The two groups represent a group on the north and one on the south of the AOP (Figure 2C). Some individuals are totally assigned to the north or south group, but the STRUCTURE analysis also detected admixed individuals (FLMNH 34910, FLMNH 32103, FLMNH 3982, FLMNH 1409b, CCDB 6105, CCDB 6130a, CCDB 6130f, CCDB 1537b, CCDB 353). Admixed individuals are not restricted to a specific geographical location. When we combine the results coming from ddRAD data and mtDNA haplotypes, we also see evidence of mitonuclear discordance. For instance, in the STRUCTURE plot, an individual from Saint Martin (FLMNH 32103) has the south mtDNA, but it has the genetic signatures of both the south and north. In contrast, individuals from Florida (FLMNH 26242, FLMNH 1476) are assigned to the north group based on mtDNA but are assigned to the south cluster under the ddRAD analysis (Figure 2B and 2C). The pairwise F_ST_ indicates high divergence between genetic groups detected by STRUCTURE (F_ST_ = 0.267). F_ST_ across the genome indicates moderate to high differentiation in all chromosomes (Figure 2D).

### 3.5. Divergence Time, Migration, and Effective Population Size

GADMA indicates that the most probable demographic scenario is allopatric divergence followed by secondary contact at different times (log-likelihood = −260.37; AIC score = 548.74) (Figure 3). The split between the north and south groups occurred at approximately ~8 Mya (Genetic Units = 5.09). The north group shows a linear population size increase until ~150 thousand years ago (Kya) (Genetic Units = 0.09) when it shows a decrease. The south group shows a constant population size until 150 Kya when it shows signals of population expansion. After a period of no contact between the two genetic groups, GADMA detected migration from north to south at approximately ~4 Mya and bidirectional migration starting at ~150 Kya. Migration from south to north is higher than from north to south, indicating asymmetrical migration.

## 4. Discussion

Our results indicate allopatric divergence following the emergence of the Amazon–Orinoco plume (AOP) between *C. ornatus* on the north and south sides of the AOP. However, we also found evidence for secondary contact, the first occurring during the Miocene (~4 Mya) and characterized by migration from the south to the north, and the second occurring during the Pleistocene and characterized by asymmetrical migration, which causes admixture between both lineages. These results challenge the validity of *C. ornatus* as a single species and suggest a cryptic speciation scenario with gene flow. Employing mtDNA and ddRAD-seq, we provide an example of how dynamic the diversification process can be in the marine realm and stress the importance of the AOP for the diversification of marine species within this region.

### 4.1. The Role of the Amazon–Orinoco Plume (AOP) in the Diversification of Callinectes ornatus

We found high genetic differentiation between groups separated by the AOP (north and south) in both types of markers (mtDNA and SNPs). Here, we added to [25] COI dataset sequences from three other locations: the north coast of Brazil, the northern coast of South America (within the plume), and the North Carolina samples. As expected, the North Carolina mtDNA haplotypes fall within the north network, but sequences from within the plume are represented by the most common haplotype in the south network. This pattern can be explained by the North Brazil Current (NBC—[73,74]) transporting individuals in the northward direction (south to north—see Figure 2A). Uniparental inheritance, smaller effective population size, and high mutation rate in comparison to nuDNA [75,76] in combination with the AOP acting as a barrier can explain the deep split we see between north and south despite continuous distribution. When looking at ddRAD markers, pairwise F_ST_ and STRUCTURE also find two genetic groups (north and south), showing high genetic differentiation (F_ST_ = 0.26). However, the STRUCTURE plot also shows admixture between north and south, indicating that complete differentiation has not occurred, as expected in cases of complete speciation (e.g., [77,78]). We find individuals being assigned to the opposite cluster, despite their geographic location and individuals showing signs of mixture and mitonuclear discordance. Admixed individuals are found in Florida and Saint Martin and also in the northeastern and southeastern coast of Brazil; mitonuclear discordance was only found in individuals on the north side of the AOP.

Considering that there are sister species separated by the AOP in a broad range of taxa (e.g., [22]—mussel, [79]—fish), our scenario could be explained by complete isolation but not complete divergence due to large effective population sizes reducing genetic drift [80]. We modeled the most probable demographic scenario to gain insight into the mechanisms that may lead to our results. However, our most probable scenario indicates divergence followed by secondary contact at different times. The estimated divergence time between the north and south groups is approximately 8 Mya, but then we detect a weak signal of migration from south to north around 4 Mya and a following phase of asymmetric migration. The geological formation of the AOP is characterized by different events, with the initial formation at 8–10 Mya [15]. A meta-analysis investigating the divergence time of reef-fish species along the Western Atlantic shows that this is the moment in which many species started to diverge, reinforcing the role of the AOP in the diversification of marine species from this region [20]. It is still unclear which traits play a role in how species respond to the barrier, but data on fish show that larger species seem to be less affected by the barrier [20]. This is probably because size seems to be associated with the ability to cross soft barriers, long-distance migration, and salinity and turbid water resistance [81]. Based on the species distribution data of all major animal groups on the north and south sides of the AOP, low salinity is evoked as one of the major factors contributing to the hampering dispersal [82]. In our case, salinity tolerance can be the trait responsible for species diverging at this initial phase of the AOP formation considering available data on *C. ornatus’s* sensitivity to salinity changes. A comparative phylogeographic study investigating *C. ornatus* and *C. danae*, two swimming crabs with the same dispersal potential but different salinity tolerance, indicates that the tolerant species (*C. danae*) is not affected by the plume [25]. Therefore, the onset of the AOP might be the kickstart of the divergence between *C. ornatus* groups on the north and south sides of the plume.

GADMA detected low migration rates from south to north around 4 Mya, which can be explained by the closure of the Isthmus of Panama, leading to the formation of the North Brazil Current (NBC) [83]. However, this migration ceased, probably because of the progression of the Andean uplift, which resulted in increased freshwater and sediment outflow around the same time [15]. The isolation between the two groups onwards can also be explained by the sea level changes throughout the Pleistocene, causing decreases in coastal habitats and closing deep reefs below the plume that could be used as steppingstones [84]. However, our demographic model also indicates another signal of migration around 150 Kya, characterized by an asymmetric migration (south to north > north to south). Secondary contact after a period of isolation can also be explained by the changing cycles during the Pleistocene [84]. For instance, although coastal areas decreased during sea level falls, they also increased during the next moment of sea level rise [84]. Our demographic model shows population expansion of the north around 150 Kya, which might have increased the chances of contact between the north and south. This secondary contact during this time can also be explained by changes that occurred during the Pleistocene, such as changes in marine currents’ dynamic [15,85,86,87]. The NBC retroflection (i.e., the event when the NBC flows southwards and west) is a common phenomenon from the Pleistocene until the present, which might intensify the transport of larvae out of the AOP influence and make the contact between south and north more frequent [88]. Therefore, our results indicate that the AOP represents a barrier for *C. ornatus*, but variations in the plume over time might open opportunities for larvae to be transported between regions, especially from south to north, because of the NBC flowing northwards. Even with two moments of secondary contact (4 Mya and present), the allopatric periods between north and south might have been enough to accumulate divergence caused by genetic drift. This result is corroborated by the fact that we found regions showing high F_ST_ across the whole genome and not concentrated in specific regions, indicating that genome-wide divergence has been accumulating for a long time [89]. At the same time, a long PLD coupled with the plume and current dynamics might result in occasional gene flow between north and south groups. The diversification process in marine species should be seen not only considering potential barriers but also the likelihood of larvae overcoming these barriers over time. Therefore, multiple scenarios are possible, specifically in marine systems [90].

### 4.2. Speciation with Gene Flow in Callinectes ornatus

A traditional paradigm postulates marine environments as open systems because of the lack of barriers and potential broad dispersal due to an extended larval stage of several marine species [91]. However, empirical studies reveal diversification occurring without complete isolation and cryptic speciation as a common phenomenon at sea [5,92]. In fact, diversification is characterized by not only strict isolation but also isolation with migration, ancient migration, secondary contact, or periodic connectivity [90]. The interplay between semi-permeable barriers and dispersal potential results in speciation with gene flow as a likely scenario in marine species [11], which also seems to be the case for *C. ornatus*.

Even isolated for millions of years, lineages can interbreed after secondary contact if no reproduction isolation has evolved [89,93]. As we discussed, larvae might occasionally travel across the AOP and promote hybridization between groups. Potential hybrids (i.e., individuals showing admixture) were found in Florida, Saint Martin, and Brazil, indicating no hybrid zone or unique region of contact between the genetic groups. Higher migration in the northward direction might explain the occurrence of mitonuclear discordance only on the north side of the AOP. The system might be characterized as a population–species continuum throughout the species distribution [94,95]. This is in accordance with the fact that adults are not migratory, but larvae can reach further regions. Based on genetic and distributional data, previous studies showed that the connection between both sides of the AOP is always in the northward direction [20,82]. Using more regions across the genome and modeling the demographic history, our results expand this current vision and provide evidence of asymmetric migration. The occurrence of admixtured individuals and mitonuclear discordance on both sides of the plume indicate that hybridization has been occurring since the south and north came in contact again, reinforcing connectivity in both directions.

Speciation is often a continuous process that eventually results in complete reproductive isolation [96]. In marine systems, we are finding speciation with gene flow increasingly common [11,97]. Under a general model of speciation with gene flow, the first phase represents positive selection on a few genes, while most of the genome shows low differentiation [89]. Later, divergence hitchhiking creates “islands of differentiation,” and, eventually, genome hitchhiking leads to great differentiation across the whole genome [89,98]. In our case, genomic differentiation across the genome shows multiple highly divergent regions and can be considered an early/intermediate moment in the speciation process [89]. We could not detect any regions under selection, which can be explained by small sample sizes or the proportion of the genome sampled, but it can also represent that genetic drift plays a role in accumulating differences between genomes over a long period of no contact. Therefore, not only does the mtDNA show levels of differentiation compatible with different species (>4%, [25]), but the whole genome seems to show the same pattern, indicating undocumented speciation within *C. ornatus*.

## 5. Conclusions

Many questions remain concerning the diversification patterns and processes of marine species [1,97]. Our work contributes to the field by employing mtDNA and ddRAD-seq to investigate the interplay between PLD and a marine barrier (AOP). Moreover, our work expands current knowledge by having an arthropod as a model since three-quarters of marine diversification studies are on Chordata, Mollusca, and Cnidaria [90]. We find evidence that the swimming crab *C. ornatus* has gone through a divergence period likely driven by the emergence of the AOP, but the two lineages mixed after secondary contact caused by larvae transported by marine currents. Low salinity tolerance is likely the trait responsible for differentiation, and PLD is responsible for secondary contact. A thorough morphological investigation has failed to reveal a diagnostic character that allows species-level differentiation ([99], personal communication). Here, we take a conservative approach to not nominating a new species due to the lack of operational criteria [100]. However, our findings provide evidence for a case of speciation with gene flow, revealing the presence of two separately evolving lineages [100].

## Figures and Tables

**Figure 1 genes-13-02263-f001:**
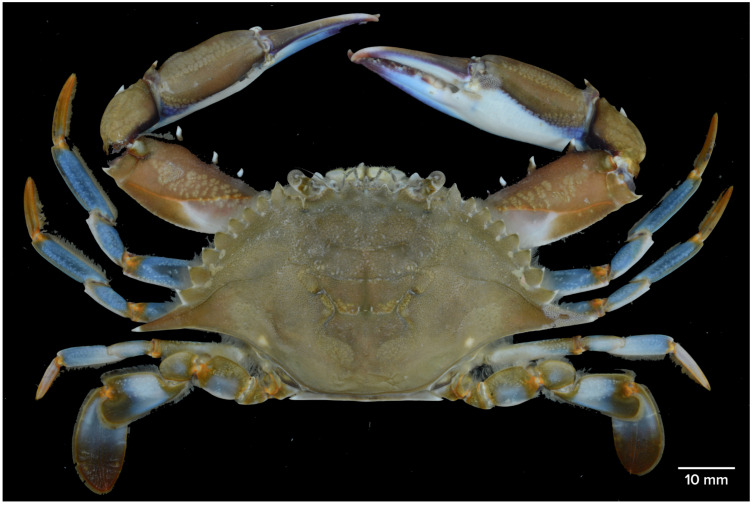
Adult male of *Callinectes ornatus* Ordway, 1863 (ULLZ 15990). Photo by DL Felder.

**Figure 2 genes-13-02263-f002:**
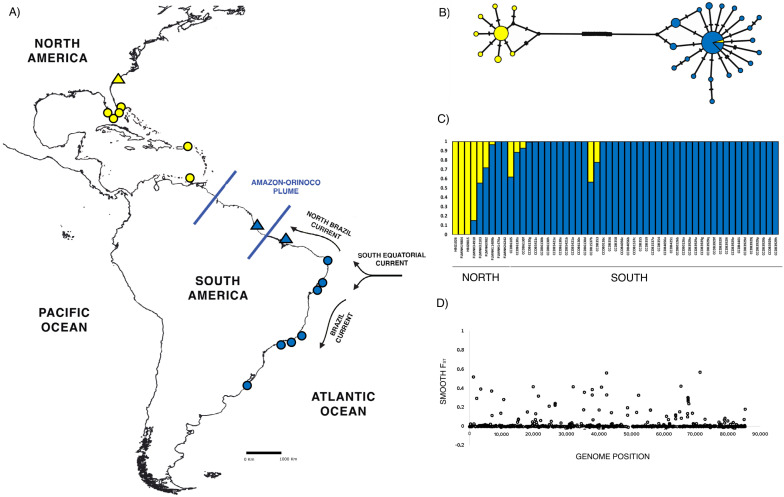
(**A**) Map of the western Atlantic showing *Callinectes ornatus* sampling localities. Circles represent the localities used for ddRAD-seq analyses; triangles represent COI new sequences added to [25]’s dataset. Yellow: north group; blue: south group. Blue lines represent the Amazon River and Orinoco River mouth, resulting in the Amazon–Orinoco Plume. Arrows represent oceanic currents; (**B**) haplotype network result for *C. ornatus*. The size of the network circles is proportional to the haplotype frequency. Crossed lines indicate the number of substitutions between haplotypes. Colors represent the region from where the haplotype was sampled. North group: yellow; south group: blue. Individuals from within the AOP region are represented in a separate section in the central haplotype of the south group. Twenty substitutions separate the two networks (**C**) STRUCTURE plot resulting from the analysis of *C. ornatus* individuals. Each vertical bar represents an individual. Different colors represent different genetic clusters (K). The voucher numbers of each individual and geographic location are indicated under the plot. (**D**) Smooth F_ST_ across the genome by calculating kernel-smoothed F_ST_ considering a sliding window of 900 Kbp. Each circle represents a genome region.

**Figure 3 genes-13-02263-f003:**
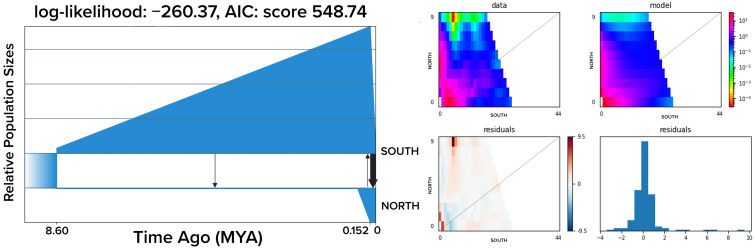
GADMA model selected (**left**) and model validation (**right**). On the graph depicting the demographic model, arrows represent migration direction, the arrow’s thickness represents migration intensity, and blue figures represent genetic groups over time. Graphs depicting model validation indicate a good fit between the simulated demographic model and our data and indicate residuals near zero.

**Table 1 genes-13-02263-t001:** Summary statistics of *Callinectes ornatus* genetic groups from the north and south sides of the Amazon–Orinoco Plume (AOP). Values are based on the analysis of 899 SNPs from across the genome. Geographic location: indicates the number of individuals collected on the north or south side of the plume and included in the analyses after all filtering steps. N: number of individuals after all filtering steps. H_E_: expected heterozygosity. H_O_: observed heterozygosity. G_IS_: inbreeding coefficient.

Geographic Location	N	H_E_	H_O_	G_IS_
North	9	0.282	0.148	0.282
South	44	0.236	0.165	0.236

## Data Availability

Demultiplexed sequence reads (forward and reverse) for every specimen are deposited in SRA (BioProject PRJNA901448). The COI haplotype data generated and used in this study are deposited in NCBI Nucleotide Database (Appendix A).

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
