# Peer review of "Genomic Analyses Implicate the Amazon–Orinoco Plume as the Driver of Cryptic Speciation in a Swimming Crab"

_genes, 2022, doi:10.3390/genes13122263_

Round 1
Reviewer 1 Report
Congratulations to the authors on this nice work – the text is well-written and easy to read. I thought the premise was interesting and the analyses were appropriate. It would have been nice to see additional analyses performed with the SNP data beyond structure and gadma to add additional strength to the cryptic speciation argument - like comparing structure to DAPC/PCA and more importantly, a phylogenetic analysis (perhaps splitstree or treemix or similar?). I’d also like a brief mention of what the strengths and weaknesses of gadma are, since most of the argument relies on this analysis. Additional line comments are below – most importantly, please improve the readability and interpretation of figures 2 and 3.
Line edits:
Abstract:
Line. 18. Please italicize C. ornatus
Line 19. North and south don’t need to be capitalized in this context.
Line 23. I recommend “long larval dispersal” rather than dispersion
Introduction:
Line 33. Please change to “results in”
Line 85. It looks like sometimes there are two spaces after periods and sometimes one, please make sure to put only a single space after periods.
Line 96. Please italicize C. ornatus throughout; however, to make it easier to read, I recommend using the common name if possible instead for subsequent references to the species.
Materials and Methods:
Line 174. Please italicize Callinectes sapidus
Line 210. Did you use location priors in the Structure run?
Line 235. Please edit typo, “mutation rated”
Results:
Line 246. I think this is the first time you refer to the southern group as the “BR group”. “BR group” isn’t mentioned in the figure, so please either call it that earlier, or keep the same label throughout (consistently use South Group instead of Brazil Group).
Line 269. Although it’s obvious that N is North and S is South, I would personally prefer if you stick with one way of referring to the groups throughout. If you want to use North and South, just at some point show that you will use N and S thereafter.
Line 272. When you refer to the Florida samples with different mitochondrial and nuclear cluster membership, it would be helpful if the reader could see that in the figure – consider perhaps labeling the structure plot with collection localities.
Line 273. How did you define North and South for the Fst analysis in Genodive? In the text you refer to the Fst between the groups detected by Structure – how did you assign the individuals that were admixed in this analysis?
Discussion:
Line 320. Please add more to this discussion of admixture and individuals being assigned to clusters that don’t correspond to their geographic localities. In the results, you discuss only one admixed individual from St. Martin and two individuals from Florida that are assigned to the S nuDNA group. However there are more individuals with apparent mixed ancestry in the plot in Fig. 2. Please discuss, and I think making the labels more readable and reflecting collection locality rather than voucher/accession number will be helpful. You could make a supplementary table also indicating each individual, their haplotype assignment, and their SNP group assignment.
Line 364. I recommend rewording “till nowadays” to “until the present” or something similar.
Line 396. Please consider rewording this phrase about the connection being always in the northward direction. Maybe something like “Previous studies showed…” or similar.
Line 399. Please edit this typo, “admixture individual” to “admixed individuals”.
Figures:
1. How big is the crab? I would love to see a scale bar! Or, maybe approx. dimensions?
2B. How many bps separate the two haplogroups? Maybe you could mention this in the caption, or write it in the figure itself.
2C. The individual labels are not readable in this figure – please either make them larger/clearer and also indicate collection locality, or just indicate the region the samples are from in this figure. The reader would benefit from being able to see the Florida samples, for example. Also the line between North and South is too thin to see and we can’t see the break between them. Please make this easier to differentiate.
2D. Please do not refer to the text for the description of this figure. The description on line 217 is short enough to fit in the caption, and it makes it easier for the reader to interpret the figure. Please indicate the size of the sliding window here in the caption.
Figure 3. I think the demographic model chosen by GADMA is enough to show, and the other figures could go in the supplement. If you leave them in the main text, maybe consider explaining them a bit more. I also think you should not so prominently show the log likelihood or AIC score since they have no meaning in isolation.
Reviewer 2 Report
The paper examines drivers of diversification in swimming crab, Callinectes ornatus populations across the Western Atlantic. Using mtDNA sequences and SNP markers from ddRADseq revealed broadly concordant patterns of genetic differentiation between populations north and south of the Amazon-Orinoco Plume, suggesting cryptic speciation due to allopatric divergence followed by periods of secondary contact, with some signs of hybridization. The paper's objectives are clearly stated, the methods are appropriate and comprehensive, and the conclusions are well-supported by the data/results. The paper is clearly written. I have no further comments.
